# How Robust Are Energy-Based Models Trained With Equilibrium Propagation?

## Abstract

Deep neural networks (DNNs) are easily fooled by adversarial perturbations that are imperceptible to humans. Adversarial training, a process where adversarial examples are added to the training set, is the current state-of-the-art defense against adversarial attacks, but it lowers the model's accuracy on clean inputs, is computationally expensive, and offers less robustness to natural noise. In contrast, energy-based models (EBMs), which were designed for efficient implementation in neuromorphic hardware and physical systems, incorporate feedback connections from each layer to the previous layer, yielding a recurrent, deep-attractor architecture which we hypothesize should make them naturally robust. Our work is the first to explore the robustness of EBMs to both natural corruptions and adversarial attacks, which we do using the CIFAR-10 and CIFAR-100 datasets. We demonstrate that EBMs are more robust than transformers and display comparable robustness to adversarially-trained DNNs on gradient-based (white-box) attacks, query-based (black-box) attacks, and natural perturbations without sacrificing clean accuracy, and without the need for adversarial training or additional training techniques.

## 1 Introduction

Deep neural networks (DNNs) are easily fooled by carefully crafted perturbations (i.e., adversarial attacks) that are imperceptible to humans Szegedy et al. (2014); Carlini & Wagner (2017); Madry et al. (2017), as well as natural noise Hendrycks & Dietterich (2019). Adversarial training, a process which involves training on adversarial examples, is the current state-of-the-art defense against adversarial attacks Madry et al. (2017). However, adversarial training is computationally expensive and also leads to a drop in accuracy on clean/unperturbed test data Tsipras et al. (2018), a well-established tradeoff that has been described theoretically Schmidt et al. (2018); Zhang et al. (2019) and observed experimentally Stutz et al. (2019); Raghunathan et al. (2019). Moreover, adversarially-trained models overfit to the attack they are trained with and perform poorly under different attacks Wang et al. (2020), as well as natural noise/corruptions. On the other hand, ViTs have shown increased robustness compared to standard Convolutional Neural Networks (CNNs) without requiring adversarial training. However, ViTs require very large datasets containing millions of samples or more to achieve good clean and robust accuracy Lee et al. (2022), which is simply not realistic in many applications.

In contrast, biological perceptual systems are much more robust to noise and perturbations, can learn from much fewer examples, and require much less power than standard DNNs. One reason for this discrepancy in performance/behavior is the fact that DNNs lack many well-known structural motifs present in biological sensory systems. For example, feedback connections are abundant in virtually every sensory area of mammals Erişir et al. (1997); Ghazanfar et al. (2001); Boyd et al. (2012); Homma et al. (2017); Jin et al. (2021), often outnumbering feedforward connections by many times Van Essen & Maunsell (1983); Erişir et al. (1997), yet they remain absent in the vast majority of DNNs. Ample neuroscientific evidence suggests that these massive feedback networks convey rich information from higher to lower cortical areas, such as sensory context, Angelucci & Bressloff (2006); Czigler & Winkler (2010); Angelucci et al. (2017), top-down attention Luck et al. (1997); Noudoost et al. (2010), and expectation Rao & Ballard (1999). It is also thought that feedback is critical for reliable inference from weak or noisy stimuli DiCarlo et al. (2012), especially

in real-world or complex scenarios with competing stimuli Desimone & Duncan (1995); Kastner & Ungerleider (2001); McMains & Kastner (2011); Homma et al. (2017).

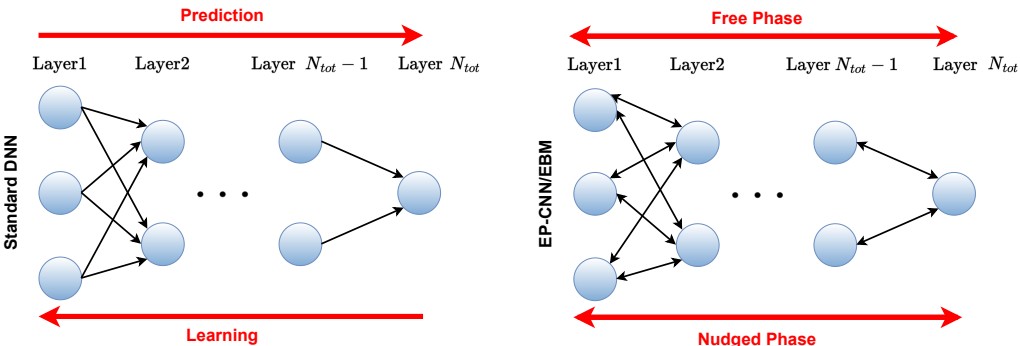

Figure 1: Standard DNNs (left) vs. EP-CNNs (right). In DNNs, learning occurs by propagating errors backwards, while inference involves only forward connections. In EP-CNNs, learning (nudged phase) as well as inference (free phase) are dynamic recurrent processes where information travels bidirectionally through feedforward and feedback connections.

Motivated by this, we hypothesize that, when incorporated into standard DNNs, top-down feedback will lead to increased robustness against adversarial attacks and natural corruptions on standard image recognition tasks. To investigate this, we focus on a recent class of biologically-plausible DNNs referred to as Energy-Based Models (EBMs), which are trained with a learning framework referred to as Equilibrium Propagation (EP). In contrast to standard DNNs, information in EBMs flows both forward and backward due to the incorporation of feedback connections between consecutive layers. These feedback connections allow EBMs to be trained with a spatio-temporally local update rule (EP) Scellier & Bengio (2017); Luczak et al. (2022) in low-power neuromorphic hardware, an important factor given the environmental cost of training DNNs on standard hardware Strubell et al. (2020); Van Wynsberghe (2021). This feedback also endows EBMs with global attractors, which should make EBMs more robust to perturbations.

After the landmark study by Scellier & Bengio (2017), recent advances in EP have focused primarily on scaling EBMs to larger and more complex tasks Laborieux et al. (2021); Kubo et al. (2022) or modifying them for use in non-standard hardware Kendall et al. (2020); Laborieux & Zenke (2022). As a result, studies on the robustness of EP-based models to adversarial and natural perturbations remain nonexistent.

Here, we perform the first investigation on the robustness of EBMs trained with EP, which we refer to as **EP-CNNs**, to adversarial and natural perturbations. Based on this investigation, we show that EP-CNNs:

- are significantly more robust than standard DNNs on white-box (where the adversary has complete information of input features, model architecture and trained weights) and black-box (adversary only has access to input features and model predictions) adversarial attacks without losing accuracy on clean test data.
- are also robust to natural perturbations, unlike adversarially-trained DNNs which perform poorly under natural perturbations.
- lead to semantic adversarial perturbations, unlike standard and adversarially-trained DNNs.

## 2 BACKGROUND AND RELATED WORKS

### 2.1 BIO-INSPIRED ADVERSARIAL DEFENSES

The use of recurrent networks that involve complex dynamics to reach a steady state is common in biologically plausible defense methods. Work by Tadros et al. (2019) has shown that sleep reactivation or replay extracts the gist of the training data without relying too much on a specific training data set. This serves a dual purpose of preventing catastrophic forgetting González et al. (2020);

Golden et al. (2022) as well as increasing adversarial robustness Chen et al. (2021); Tadros et al. (2022). Inclusion of trainable feedback connections inspired by regions in cerebral cortex Walsh et al. (2020), in order to implement predicting coding frameworks, have demonstrated marginal robustness against both black-box and white-box attacks Boutin et al. (2020); Choksi et al. (2021). Evidence of perceptual straightening of natural movie sequences in human visual perception Hénaff et al. (2019) has also inspired robust perceptual DNNs which integrate visual information over time, leading to robust image classification Vargas et al. (2020); Daniali & Kim (2023). However, the above models have either not been tested against adversarial or natural perturbations, and the ones that have, have only exhibited marginal increases in robustness relative to standard DNNs.

## 2.2 EQUILIBRIUM MODELS

A line of work similar to equilibrium propagation was introduced by Bai et al. (2019), known as deep equilibrium models (DEQ). DEQs involve finding fixed points of a single layer and since the fixed point can be thought of as a local attractor, these models were expected to be robust to small input perturbations, although empirical observations have proven otherwise Gurumurthy et al. (2021). The process of finding this fixed point usually involves a black-box solver e.g., Broyden's method, which is beneficial in terms of memory usage. For robustness evaluations however, previous studies have used approximate/inexact gradients in order to carry out gradient-based attacks, raising concerns about gradient obfuscation Liang et al. (2021); Wei & Kolter (2021); Yang et al. (2022). Even so, DEQs alone are not robust to adversarial attacks, and, as a result, are often paired with adversarial training or other additional techniques to gain robustness Li et al. (2022); Chu et al. (2023); Yang et al. (2023).

## 2.3 LEARNING IN PHYSICAL SYSTEMS

With an ever-increasing shift towards spiking neural networks (SNNs) from traditional deep neural networks (DNNs), there is a greater need to test these models for robustness. Training an SNN with backpropagation has been shown to be inherently robust to gradient-based attacks conducted on hardware when compared to similar attacks on DNNs Marchisio et al. (2020); Sharmin et al. (2020).

The inherent robustness of learning in physical systems in general could be attributed to

1. Local learning rules where the update of each layer's weights are dependent on its previous and next layers' states unlike backpropagation which involves computing and storage of error computation graphs.

2. The complex energy landscape that makes storage of memories possible. Since energy minimization in these physical systems is directly performed by the laws of physics Stern et al. (2021); Stern & Murugan (2023) and not with numerical methods in a computer simulation, the expectation is that any gradient-based attack would involve crossing a decision boundary that changes the input semantically Lavrentovich et al. (2017).

## 3 METHODS

### 3.1 EXISTING EQUILIBRIUM PROPAGATION FRAMEWORK

EP, as introduced by Scellier & Bengio (2017) makes use of recurrent dynamics, where input $x$ to the system is held static and the state $s$ of the neural network converges to a steady state $s_*$. In the deep ConvNets implementation of EP Laborieux et al. (2021), the Hopfield-like energy function for a neural network with $N_{\text{conv}}$ convolutional layers and $N_{\text{tot}}$ being the total number of layers, is described by

$$\Phi(x, \{s^n\}) = \sum_{n < N_{\text{conv}}} s^{n+1} \cdot \mathcal{P}(w_{n+1} \star s^n) + \sum_{n=N_{\text{conv}}}^{N_{\text{tot}}-1} s^{n+1^\intercal} \cdot w_{n+1} \cdot s^n \tag{1}$$

where $s^n$ is the state of the $n^{th}$ layer with $s^0$ being equal to the input $x$, $w_{n+1}$ are the convolutional/linear weights connecting states $s^n$ to $s^{n+1}$ and $\mathcal{P}$ being a pooling operation following the

convolution. The time-evolution of the state $s$ is then given as

$$s_{t+1} = \frac{\partial \Phi}{\partial s_t} \tag{2}$$

where convergence to the steady state $s_*$ is attained when

$$s_* = \frac{\partial \Phi}{\partial s_*} \tag{3}$$

We trained the EP models using a symmetric weight update rule as described in Laborieux et al. (2021). Details of the training rules are presented in appendix A. After obtaining a model trained with EP, one then evolves the network under the recurrent dynamics as given by Equation 2. During this time evolution, also referred to as the free phase in literature, it is possible to use the state of the last layer $s^{N_{tot}}$ to make predictions. One could observe the accuracy of these predictions as a function of the free phase iterations $t$, as shown in Figure 4a. The accuracy of the model improves until the steady state (Equation 3) is reached, after which, the accuracy does not increase with iterations, see Figure 4a. We also observe that the accuracy stays at $10\%$ for the first three timesteps since the trained model we used consists of four layers and the last layer is not updated until the fourth timestep, see Equation 2. In the next subsection, we describe how we performed gradient-based attacks by making use of these time-dependent predictions.

## 3.2 TEMPORAL PROJECTED GRADIENT DESCENT ATTACKS

First introduced by Madry et al. (2017), Projected Gradient Descent (PGD) attack outlines an iterative procedure for finding adversarial examples when the adversary has access to underlying gradients. In this section we show the convergence properties of PGD attacks on networks trained with EP. Let $S$ be a $l_p$-sphere of radius $\epsilon$ around the unperturbed image $x$. The attacks start at a random point $x^0 \in S$, and follow the iterations:

$$x^{i+1} = \Pi_S(x^i + \alpha \cdot g^i) \tag{4}$$
$$\text{where } g^i = \arg\max_{\|v\|_p \leq 1} v^\intercal \nabla x^i \mathcal{L}(\hat{y}(x^i), y)$$

Here $\mathcal{L}$ is a suitable loss-function, $\hat{y}(x^i)$ is the output of the network for input $x^i$, $\alpha$ is a step-size, $\Pi_s$ is the projection of the input onto the norm-ball $S$ and $g^i$ is the steepest ascent direction for a given $l_p$-norm.
In the case of EP, a prediction $y$ could be made at any stage of the free-phase iterations i.e., $\hat{y}_{t+1} = \frac{\partial \Phi}{\partial \hat{y}_t}(x, h_t, \hat{y}_t, \theta)$. Thus, the steepest ascent direction for a given PGD iteration would be a time-dependent term in the following manner.

$$g_t^i = \arg\max_{\|v\|_p \leq 1} v^\intercal \nabla_x^i \mathcal{L}(\hat{y}_t(x^i), y)$$

$\nabla_x \mathcal{L}(\hat{y}_t(x), y)$ could be further expanded as

$$\nabla_x \mathcal{L}(\hat{y}_t(x), y) = \frac{\partial \mathcal{L}}{\partial \hat{y}_t} \frac{\partial \hat{y}_t}{\partial x} \tag{5}$$

With increasing timesteps (free phase iterations), the accuracy of a model trained with EP increases until it saturates, which implies that

$$\left. \frac{\partial \mathcal{L}(\hat{y}_t(x^i), y)}{\partial \hat{y}_t} \right|_{t>T} = \left. \frac{\partial \mathcal{L}(\hat{y}_t(x^i), y)}{\partial \hat{y}_t} \right|_{t=T} \tag{6}$$

where $T$ is the number of timesteps required by a model trained with EP to reach a steady state. While the second term in Equation 5 might be difficult to evaluate, Equation 6 shows that for attacks on EP that has been evolved for $t$ timesteps where $t \geq T$, the gradient $g_t$ would be saturated and cannot be maximized further. This conclusion is further confirmed by Figure 4b which shows the test accuracy of EP for a PGD attack, with the attack's success saturating at 70 timesteps.

### 3.3 ADVERSARIAL EXAMPLES

Adversarial examples involve model-specific manipulations of input data, the nature of which, is imperceptible to humans but causes classification errors with state-of-the-art neural networks trained with backpropagation Szegedy et al. (2014). The procedure for generating these adversarial examples can be split into two classes, depending on the amount of information the adversary has access to.

**White-Box Attacks** White-box attacks are gradient-based attacks where the adversary has access to the features of the input, the trained model's architecture as well as model weights. While there exist a host of such attacks, two have held their status of being state-of-the-art, C&W attacks developed by Carlini & Wagner (2017) and projected gradient descent (PGD) attacks developed by Madry et al. (2017). Our white-box attack experiments would be focused on both PGD and C&W attacks carried out on different models.

**Black-Box Attacks** Black-box attacks are inference-based attacks where the adversary does not have access to model weights and can only refine its attacks iteratively by querying the trained model multiple times. Evaluating against a black-box attack is useful to allay concerns about gradient obfuscation since it has been shown that most state-of-the-art defense methods based on white-box attacks could provide a false sense of security Athalye et al. (2018). Among a variety of black-box attacks, we chose to evaluate the robustness of our trained models against Square Attack, which is a query-efficient black box attack Andriushchenko et al. (2020). Additionally, we also performed the Auto Attack Croce & Hein (2020), a state-of-the-art attack composed of two different kinds of AutoPGD attacks, a DeepFool attack as well as a Square attack.

## 4 EXPERIMENTAL SETUP

We perform experiments on image recognition datasets using common image corruptions and adversarial attacks to test the robustness of EP-CNN. For baselines, we compare against standard CNN models, adversarially trained CNN models, and vision transformer models. Abovementioned models were averaged over five different seeds to determine errors in accuracy of the results of robustness. In addition, we show convergence properties of EP-CNN attacked at different timesteps of inference. The code we used in the following experiments can found here: https://anonymous.4open.science/r/AdvEP-F4D0.

### 4.1 MODELS

**EP-CNN** To train an EP-CNN and test it against adversarial and natural corruptions, we developed the AdvEP package which is based on the implementation by Laborieux et al. (2021) and consists of an optimized weight update rule. All of the following results we present are obtained over five random seeds (which were consistent across models). Supplementary Table 2 shows the hyperparameters used to train EP-CNN on CIFAR10 and CIFAR100 datasets. For both datasets, the model consists of 4 convolutional layers followed by a fully connected (FC) layer. The FC layer is not a part of the EP dynamics and resembles a readout layer in the case of reservoir computing.

The AdvEP package offers optimized versions of several weight update rules as provided in the supplementary material section of Laborieux et al. (2021). Among different rules, the symmetric weight update rule as described by Equation 9 performed the best under attacks. Therefore, all our results on EP have been trained with the symmetric weight update rule.

### 4.1.1 BASELINES

**BP-CNN** To compare the role of EP in robustness, we trained a CNN model, consisting of 4 convolutional layers followed by an FC layer, trained with backpropagation. Additionally, each convolutional layer was followed by a batch normalization layer. While these models did not achieve state-of-the-art accuracy on clean images, the purpose was to compare results from EP with an equivalent model.

**AdvCNN** This model's architecture and weight update rules are the same as that of **BP-CNN**. To train AdvCNN, we perform standard adversarial training Madry et al. (2017) with various $\|\epsilon\|_2 \in [0.05, 0.50, 0.75, 1.00]$ constraints and 200 queries using PGD attacks. We observe that

adversarial training signifcantly boosts robustness while also leading to a drop in clean accuracy, over unperturbed images. Our PGD attack results for the **AdvCNN**s' are consistent with the ones reported in Engstrom et al. (2019)

**ViT** We also trained a vision transformer which has a CIFAR10 performance comparable to other models' clean accuracy. The implemented model had 7 layers each with 12 heads and a patch size of $4 \times 4$. These architectural hyperparameters were chosen based on the clean test performance by performing a grid search.

## 4.2 TRAINING AND TESTING DETAILS

All models were implemented in PyTorch 2.0.0 on a high-performance computing node with eight NVIDIA GeForce A100 GPUs, 64 AMD Rome CPU cores, and 251 GB of memory. The dynamics of the system requires evaluating the gradient of the energy function with respect to the current state of the system. Since the energy-function is quadratic, we modified the code developed by Laborieux et al. (2021) by incorporating exact equations of motion. These modifications were then verified to settle into the same fixed point while providing a 30% speed-up, needing 10 hours to train a CIFAR-10 dataset as compared to 15 hours with the previous implementation. The input to each model consisted of static images of spatial dimensions $32 \times 32$. Input images were augmented during training with random cropping and horizontal flipping for **BP-CNN**, **AdvCNN** and **EP-CNN** models. For **ViT** models, we used the autoaugment technique Cubuk et al. (2019) which have led to state of the art results in supervised learning. All training hyperparameters, such as the batch size and learning rate have been provided in Table 2.

During testing, each image was normalized with the same parameters used for training, except there were no augmentations. For both white-box attacks and natural-corruptions, we evaluated the performance of all the models across the entire test set of CIFAR-10 and CIFAR-100. For black-box (Square) attacks Andriushchenko et al. (2020), we down-select a test set of 600 images from each dataset to use as inputs to the models.

## 5 RESULTS

Table 1: Mean Robustness Across AutoAttack, PGD, Square, and C&W Attacks for All Attack Strengths

| Model | CIFAR-10 | CIFAR-100 |
|---|---|---|
| BP-CNN | 0.43 | 0.33 |
| Adv-CNN($\epsilon = 0.5$) | 0.60 | 0.45 |
| Adv-CNN($\epsilon = 1.0$) | **0.61** | **0.46** |
| ViT | 0.51 | 0.31 |
| EP-CNN | 0.58 | 0.44 |

## 5.1 EP-CNNS ARE MORE ROBUST TO NATURAL CORRUPTIONS

**EP-CNN, BP-CNN** and **AdvCNN** models were trained with HorizontalFlip and RandomCrop transformations applied to the original images. Since ViT models were trained with AutoAugment Cubuk et al. (2019) which involve augmentations that are similar to the corruptions being presented, however the robust accuracy achieved by **ViT** did not surpass other models. **EP-CNN** models outperformed other models in all corruptions except the contrast corruption. The CIFAR-C dataset contains an array of corruptions that could affect model performance in real-world settings and hence are a necessary check for energy-based models.

## 5.2 EP-CNNS ARE INHERENTLY ROBUST TO STATE-OF-THE-ART ATTACKS

Given the differentiable **EP-CNN** model, we are able to perform exact gradient-based attacks on the models. Specifically, we used projected gradient descent (PGD) attack Madry et al. (2017) and C&W attacks Carlini & Wagner (2017). We use the Adversarial Robustness Toolbox to implement the attack with both $l_\infty$ and $l_2$ constraints on the CIFAR-10 and CIFAR-100 dataset. Across all

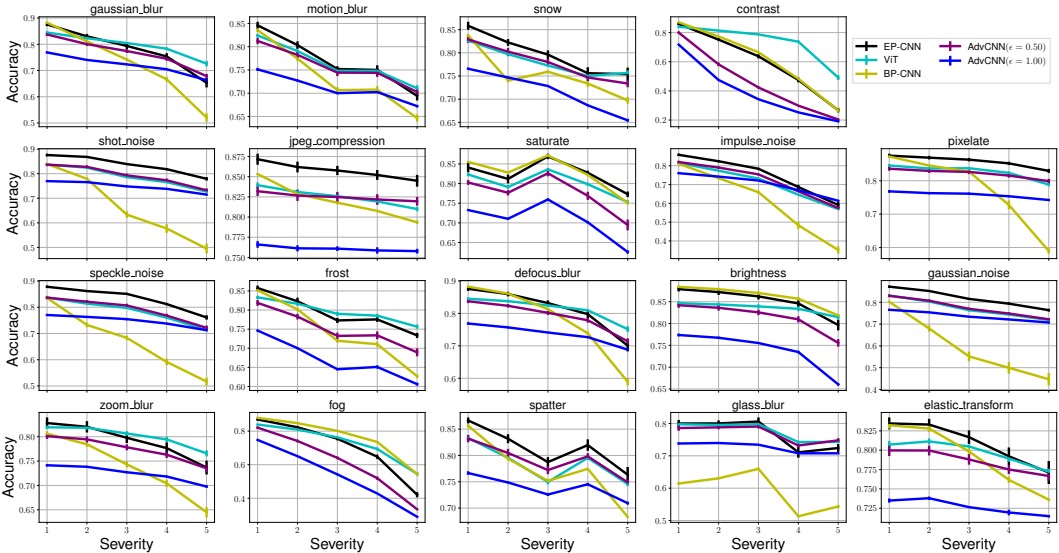

Figure 2: Line graph of accuracy under CIFAR10-C dataset. Lines: **BP-CNN**, **AdvCNN** with $l_2$ $\epsilon = 0.50, 1.00$, **EP-CNN** and **ViT**. x-axis: Corruption Severity (amount of noise added), y-axis: Accuracy. Error bars represent the 95% CI from 5 different runs/seeds.

models, we set the number of attack iterations to 20 and the step-size to $\epsilon/8$ where $\epsilon$ is the attack perturbation.

On the PGD attacks and AutoAttacks, the **AdvCNN**s outperform other models (Figures 3 and 9), in the order of highest (blue) to lowest (red) $\epsilon$ used for adversarial training. Of the models trained without adversarial training, the **EP-CNN** displayed the next best performance across both CIFAR-10 and CIFAR-100 datasets, albeit without a drop in its clean accuracy, followed by **ViT** and **BP-CNN**. While **EP-CNN** is not as robust as the adversarially trained **AdvCNN**s' under PGD attacks, it is more robust compared to the standard **BP-CNN**s' and **ViT**. By observing the PGD attacks on each model, it is evident that the attacked images computed on **EP-CNN**s have been modified in a meaningful/semantic way (Figure 5). This is in contrast to all other models tested, including the adversarially-trained CNNs and the **ViT**, and is one indication that **EP-CNN**s may learn features that are actually meaningful, as opposed to those that are spuriously predictive of the task Ilyas et al. (2019). In case of C&W attacks over the CIFAR-100 dataset, **EP-CNN** performed the best (see Figure 11a) and was observed to be as robust as the **AdvCNN**s', for C&W attacks over the CIFAR-10 dataset as shown in Figure 11b.

**Checking for gradient obfuscation.** Since **EP-CNN**s converge to an equilibrium point in the free phase (aka inference), which may cause the gradient to vanish, we took different measures to ensure that **EP-CNN**s were not using gradient obfuscation to achieve robustness Athalye et al. (2018). Since the AutoAttack implements a black-box attack, and the **EP-CNN**'s performance on AutoAttack (Figure 3a) was similar to that on the PGD attack (Figure 9b), this is one indication that the robustness of **EP-CNN**s is not due to gradient obfuscation. To further check for gradient obfuscation, we also performed PGD attacks on **EP-CNN**s using different numbers of free phase iterations. If the **EP-CNN** was performing gradient obfuscation, it should be more robust to the PGD attack when using more free phase iterations (i.e. as the **EP-CNN** is allowed to get closer to the equilibrium point). However, we observe the opposite trend, in which the **EP-CNN**'s accuracy is higher with less free phase iterations (Figure 4b). Overall, this provides strong evidence that **EP-CNN**s are actually robust, and are not relying on gradient obfuscation.

## 5.3 EP-CNNS ARE ROBUST TO HIGH-STRENGTH BLACK-BOX ATTACKS

**AdvCNN** ($\epsilon = 1.00$) and **AdvCNN** ($\epsilon = 0.50$) display the best robustness against Square attacks, although with a reduced clean accuracy, as shown in Figure 10a. **EP-CNN** displays the next best performance, exhibiting close performance to the adversarially-trained models. Surprisingly, **ViT**

performed poorly with square attacks, performing worse than the non-adversarially trained **BP-CNN**.

We evaluated the average robust accuracy across various adversarial attacks for different models and different datasets and have summarized the results in Table 1. We conclude that the robust performance of **EP-CNN**s is closer to **AdvCNN** models as compared to **BP-CNN** and **ViT**.

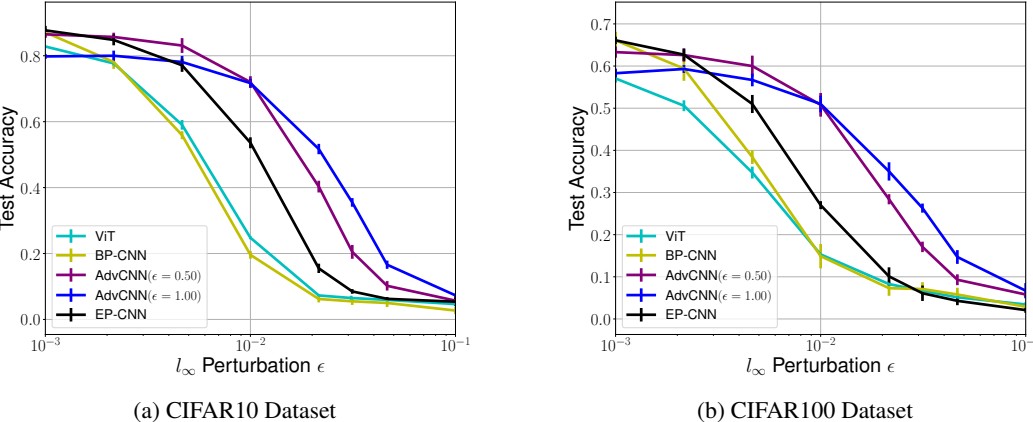

(a) CIFAR10 Dataset          (b) CIFAR100 Dataset

Figure 3: AutoAttack results on CIFAR-10 and CIFAR-100 dataset with $l_\infty$ perturbations. Lines: **BP-CNN**, **AdvCNN** with $l_2$ $\epsilon = 0.50,\ 1.00$, **EP-CNN** and **ViT**. Error bars represent the 95% CI over 5 different runs/seeds.

## 6 DISCUSSION

In this work, we evaluate the adversarial robustness of Equilibrium Propagation, a biologically plausible learning framework, compatible with neuromorphic hardware, for image classification tasks. We demonstrate clean accuracy comparable to models trained with backpropagation. Through our experiments, we demonstrate competitive accuracy and inherent adversarial robustness of **EP-CNN**s to natural corruptions and black-box attacks. We also demonstrate competitive robustness to white-box attacks when compared with adversarially-trained models.

**EP-CNN**s far outperform the ViT models across the datasets used in this study for both adversarial and natural noise, even though ViT models were trained on input extensively augmented using similar noise perturbations. **EP-CNN**s also do not suffer from lower clean accuracy unlike models that have been adversarially trained. These adversarially trained models also fail catastrophically when subjected to noise they were not trained on, such as the natural corruptions, as shown in Figures 2 and 13, whereas **EP-CNN**s are far more robust across both adversarial attacks and natural corruptions, without any extensive augmentation or adversarial training. A summary of performances of all models has been provided in Tables 3, 4, 5, 6 and an average score for all models across different attacks and different strengths, has been summarized in Table 1.

The role of feedback connections in the brain has long been overlooked in DNNs. Neuroscientific studies suggest that the abundant feedback connections present in the cortex are not merely modulatory and convey valuable information from higher to lower cortical areas, such as sensory context and top-down attention. Recent experiments have shown that time-limited humans process adversarial images much differently compared to their DNN counterparts Elsayed et al. (2018), thus leading to the hypothesis that perception of static images is a dynamic process and benefits hugely from recurrent feedback connections Daniali & Kim (2023). Earlier studies Hupé et al. (1998) hypothesized the role of feedback connections in discriminating the object of interest from background information and recent studies Kar et al. (2019) showed that challenging images took more time to be recognized compared to control images, providing more reasons to believe that feedback connections is critical to improving robustness of the ventral stream. Our findings solidify the above claim, thus paving the way for robust artificial networks that include feedback connections.

While **EP** is not the only training algorithm that involves settling into a fixed point before making inference, the complex dynamics showcased in **EP** (see Figure 1) gets rid of small input perturbations in the process of attaining a steady state. In case of white-box attacks, large perturbations computed on EBMs/EP appear *semantically meaningful* as shown in Figures 5 and 7 in contrast to all other models tested, thus strengthening the hypothesis that large perturbations are able to move the trajectory of the state past a decision boundary for models trained with EP. This is concurrent with the assumption that equilibrium propagation allows learning of features of the input dataset in a hierarchical manner, akin to the hierarchical ground state structure of a spin-glass. More theoretical insights for the increased robustness of equilibrium propagation are provided in appendix B, while we leave a detailed proof for future work.

There has been a recent focus on improving the inherent robustness of DNNs, by including motifs inspired by biological perceptual systems. Using predictive coding in pre-trained models to train feedback connections improved adversarial robustness as shown in the *Predify* framework Choksi et al. (2021); Alamia et al. (2023), however the robustness to standard attacks in these studies have only been performed for very few timesteps, which could be misleading. Lateral inhibitions, previously thought to contribute to bottom-up attention, were implemented as a frontend to DNNs in the sparse-coding inspired LCANets Teti et al. (2022), which were shown to be naturally robust to natural corruptions and black-box attacks but were vulnerable to white-box attacks. Finally, a related work Kim et al. (2020) included both lateral inhibitions and top-down feedback connections and demonstrated increased robustness to adversarial attacks, however, these attacks were not conducted on the model itself, rather transfer attacks were used. In contrast, our work shows that models trained with EP achieve competitive accuracy over an array of black-box and white-box attacks (which were generated on **EP-CNN** models) as well as natural corruptions.

LIMITATIONS AND FUTURE WORK

As mentioned in the initial paper Scellier & Bengio (2017), we also found that **EP** is relatively sensitive to the hyperparameters used to train the model as well as the seed used to initialize the weights. Since the inference is defined implicitly in terms of the input and the parameters of the model, this makes even our optimized implementation less practical for applications on traditional hardware (like GPUs). While having a model sensitive to initial parameters and also time-consuming could appear as a disadvantage, it mimics the analog learning systems and could prove useful in the future. A future study could involve coming up with more reliable and stable recurrent dynamics that would give rise to robust and safer analog circuits.

Apart from the instability of EP models shown during training, another limitation of our work is the amount of time required to perform the free phase to reach a steady state, when trained on traditional GPUs. While this limits the EP models to relatively shallow architectures and datasets when using standard hardware, EP-CNNs are ideal for implementation in neuromorphic hardware, leading to faster and more robust bio-plausible models. While spiking implementations of EP exist O'Connor et al. (2019), future work would then involve optimizing those implementations in order to run on realistic datasets like ImageNet.

# 7 CONCLUSION

We performed the first investigation into the robustness of EBMs trained with EP (EP-CNNs) to adversarial and natural perturbations/noise. Existing gradient-based attack methods were adapted to be compatible with EP-CNNs. Our results indicate that EP-CNNs are significantly more robust than standard CNNs and ViTs. We also show that EP-CNNs exhibit significantly greater robustness to natural perturbations and similar robustness to state-of-the-art black-box attacks when compared with adversarially-trained CNNs, but they do not suffer from decreased accuracy on clean data. We also find that the adversarial attacks on EBMs are more semantic than those computed on standard and adversarially-trained DNNs, which indicates that EBMs learn features that are truly useful for the classification tasks they are trained on. Overall, our work indicates that many of the problems exhibited by current DNNs, including poor energy-efficiency and robustness, can be solved by an elegant, biophysically-plausible framework for free.

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

## A  WEIGHT UPDATE RULE FOR EQUILIBRIUM PROPAGATION

EP involves two stages for training. In the first stage known as the free phase, the state of the network is allowed to evolve according to Equation 2 whereas the second stage, also called the 'nudging phase' involves adding the loss term $-\beta\mathcal{L}$ to the energy function $\Phi$ where $\beta$ is the scaling factor. The dynamics during the second phase are initialized with the fixed point of the free phase and are evolved by minimizing the modified energy function as shown below:

$$s_0^\beta = s_* \quad \text{and} \quad \forall t > 0 \qquad s_t^\beta = \frac{\partial \Phi}{\partial s_{t-1}^\beta}(x, s_{t-1}^\beta, \theta) - \beta \frac{\partial \mathcal{L}}{\partial s_{t-1}^\beta}(s_{t-1}^\beta, y) \tag{7}$$

The state of the system after the nudging phase converges to a new steady state $s_*^\beta$. The original EP learning rule proposed by Scellier & Bengio (2017) involved taking a difference between the gradient of the energy function of the fixed point of the nudged phase and the fixed point of the free phase, hence falling into the class of contrastive learning algorithms.

$$\text{where} \quad \hat{\nabla}^{\text{EP}}(\beta) = \frac{1}{1\beta}\left(\frac{\partial \Phi}{\partial \theta}(x, s_*^\beta, \theta) - \frac{\partial \Phi}{\partial \theta}(x, s_*, \theta)\right) \tag{8}$$

Laborieux et al. (2021) proposed the symmetric weight update rule :

$$\text{where} \quad \hat{\nabla}^{\text{EP}}_{\text{sym}}(\beta) = \frac{1}{2\beta}\left(\frac{\partial \Phi}{\partial \theta}(x, s_*^{\beta}, \theta) - \frac{\partial \Phi}{\partial \theta}(x, s_*^{-\beta}, \theta)\right) \tag{9}$$

Comprised of a nudging $(+\beta)$ and an anti-nudging $(-\beta)$ phase, this modification (Equation 9) in the learning rule enabled the scaling of EP to harder tasks like CIFAR10, CIFAR100 datasets. As mentioned in Laborieux et al. (2021), the symmetric weight update rule is local in space, in other words, each layer's $(w_n)$ weight update is only dependent on the previous $(s^{n-1})$ and the next $(s^{n+1})$ layer's states, hence making it suitable for implementation in neuromorphic hardware.

## B  STABILITY IN ENERGY-BASED MODELS

Energy-based models learn features through the minimization of their energy function. These learned features are stored in the form of attractors of the energy landscape, and these attractors often have a basin of attraction i.e., for a small perturbation $\epsilon$ to an input $x$, the probability that perturbed input leads to the same steady state of the network $\hat{y}_*(x')$ as the original input, is given as

$$\|x - x'\|_p < \epsilon \implies \mathbb{P}(\hat{y}_*(x) = \hat{y}_*(x')) \sim \epsilon^{\alpha} \tag{10}$$

where $\alpha > 0$ is known as the uncertainty exponent. We hypothesize that learning and inference through local rules and feedback connections leads to a larger uncertainty exponent, and eventually, provides robustness to natural corruptions and adversarial perturbations. Evaluation of the uncertainty exponent would be part of future work.

## C  HYPERPARAMETERS TABLE

Table 2: Hyperparameters used for training **EP-CNN** models

| Hyperparameter | CIFAR-10 | CIFAR-100 |
|---|---|---|
| Batch Size | 128 | 128 |
| Channel Sizes | $[128, 256, 512, 512]$ | $[128, 256, 512, 1024]$ |
| Kernel Size | $[3, 3, 3, 3]$ | |
| Paddings | $[1, 1, 1, 0]$ | |
| Initial LRs | $[25, 15, 10, 8, 5] \times 10^{-2}$ | |
| $T_{\text{free}}$ | 250 | |
| $T_{\text{nudge}}$ | 30 | |

# D    RESULT SUMMARY TABLE

Table 3: PGD Attack results for various $l_2$ perturbations across different models

| | CIFAR-10 | | | | | CIFAR-100 | | | | |
|---|---|---|---|---|---|---|---|---|---|---|
| | Clean | $\epsilon = 0.07$ | $\epsilon = 0.31$ | $\epsilon = 0.49$ | $\epsilon = 1.00$ | Clean | $\epsilon = 0.07$ | $\epsilon = 0.31$ | $\epsilon = 0.49$ | $\epsilon = 1.00$ |
| BP-CNN | 0.88±0.05 | 0.81 | 0.31 | 0.15 | 0.11 | 0.67 ±0.04 | 0.58 | 0.21 | 0.13 | 0.10 |
| Adv-CNN($\epsilon = 0.5$) | 0.82±0.05 | 0.85 | **0.79** | 0.71 | 0.35 | 0.62±0.04 | **0.61** | **0.52** | 0.43 | 0.22 |
| Adv-CNN($\epsilon = 1.0$) | 0.74±0.05 | 0.79 | 0.76 | **0.72** | **0.52** | 0.56±0.04 | 0.56 | 0.52 | **0.47** | **0.31** |
| ViT | 0.85±0.05 | 0.77 | 0.48 | 0.36 | 0.19 | 0.59±0.04 | 0.52 | 0.23 | 0.14 | 0.09 |
| EP-CNN | 0.88 ±0.05 | **0.86** | 0.63 | 0.44 | 0.15 | 0.64 ±0.04 | 0.60 | 0.37 | 0.24 | 0.11 |

Table 4: AutoAttack results for various $l_\infty$ perturbations across different models

| | CIFAR-10 | | | | | CIFAR-100 | | | | |
|---|---|---|---|---|---|---|---|---|---|---|
| | Clean | $\epsilon = \frac{1}{255}$ | $\epsilon = \frac{3}{255}$ | $\epsilon = \frac{5}{255}$ | $\epsilon = \frac{8}{255}$ | Clean | $\epsilon = \frac{1}{255}$ | $\epsilon = \frac{3}{255}$ | $\epsilon = \frac{5}{255}$ | $\epsilon = \frac{8}{255}$ |
| BP-CNN | 0.88 | 0.56 | 0.20 | 0.06 | 0.05 | 0.67 | 0.38 | 0.15 | 0.07 | 0.07 |
| Adv-CNN($\epsilon = 0.5$) | 0.82 | 0.83 | **0.72** | 0.40 | 0.21 | 0.62 | **0.60** | 0.51 | 0.28 | 0.17 |
| Adv-CNN($\epsilon = 1.0$) | 0.74 | 0.78 | 0.72 | **0.52** | **0.36** | 0.56 | 0.567 | **0.51** | **0.35** | **0.26** |
| ViT | 0.85 | 0.59 | 0.25 | 0.07 | 0.07 | 0.59 | 0.35 | 0.15 | 0.08 | 0.07 |
| EP-CNN | 0.88 | **0.86** | 0.63 | 0.44 | 0.15 | 0.64 | 0.51 | 0.27 | 0.10 | 0.06 |

Table 5: Square Attack results for various $l_\infty$ perturbations across different models

| | CIFAR-10 | | | | | CIFAR-100 | | | | |
|---|---|---|---|---|---|---|---|---|---|---|
| | Clean | $\epsilon = \frac{1}{255}$ | $\epsilon = \frac{3}{255}$ | $\epsilon = \frac{5}{255}$ | $\epsilon = \frac{8}{255}$ | Clean | $\epsilon = \frac{1}{255}$ | $\epsilon = \frac{3}{255}$ | $\epsilon = \frac{5}{255}$ | $\epsilon = \frac{8}{255}$ |
| BP-CNN | 0.88 | 0.81 | 0.62 | 0.26 | 0.13 | 0.67 | 0.60 | 0.42 | 0.18 | 0.12 |
| Adv-CNN($\epsilon = 0.5$) | 0.82 | 0.85 | **0.82** | 0.66 | 0.50 | 0.62 | 0.62 | **0.58** | 0.45 | 0.32 |
| Adv-CNN($\epsilon = 1.0$) | 0.74 | 0.79 | 0.77 | **0.67** | **0.55** | 0.56 | 0.58 | 0.55 | **0.47** | **0.38** |
| ViT | 0.85 | 0.73 | 0.50 | 0.22 | 0.15 | 0.59 | 0.46 | 0.30 | 0.15 | 0.11 |
| EP-CNN | 0.88 | **0.86** | 0.81 | 0.64 | 0.47 | 0.64 | **0.64** | 0.55 | 0.36 | 0.24 |

Table 6: C&W Attack results for various adversarial constants across all models

|  | CIFAR-10 | | | | | CIFAR-100 | | | | |
|---|---|---|---|---|---|---|---|---|---|---|
|  | Clean | c=0.005 | c=0.01 | c=0.1 | c=1.0 | Clean | c=0.005 | c=0.01 | c=0.1 | c=1.0 |
| BP-CNN | 0.88 | 0.04 | 0.01 | 0.01 | 0.0 | 0.67 | 0.14 | 0.04 | 0.02 | 0.02 |
| Adv-CNN($\epsilon = 0.5$) | 0.82 | 0.17 | 0.02 | 0.06 | 0.005 | 0.62 | 0.25 | 0.14 | 0.06 | 0.044 |
| Adv-CNN($\epsilon = 1.0$) | 0.74 | 0.44 | 0.05 | 0.07 | 0.01 | 0.56 | 0.35 | 0.24 | 0.06 | 0.05 |
| ViT | 0.85 | **0.55** | **0.45** | **0.27** | **0.21** | 0.59 | 0.22 | 0.20 | 0.16 | 0.13 |
| EP-CNN | 0.88 | 0.42 | 0.24 | 0.10 | 0.08 | 0.64 | **0.51** | **0.45** | **0.35** | **0.29** |

# E    TIME-DEPENDENT CHARACTERISTICS OF EP

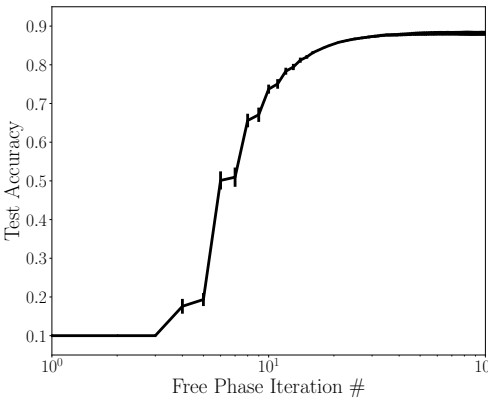
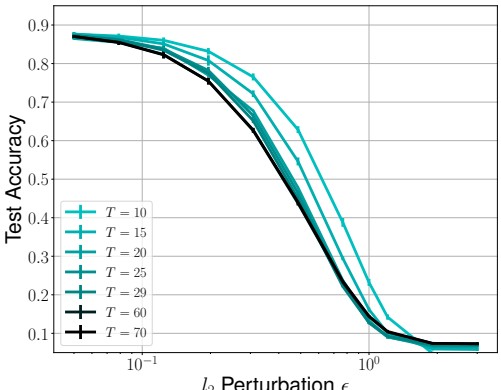

(a) Test accuracy achieved by a trained **EP-CNN** model, evaluated at each iteration of the free-phase. Error bars represent five models trained with five different seeds. With increasing number of iterations, the accuracy saturates.

(b) **EP-CNN** Accuracy vs PGD perturbation $\epsilon$ for examples generated using different timesteps, black curve representing the state of the system evolved for 10 timesteps and cyan representing attacking the state after 70 free-phase iterations. The curves for 60 and 70 timesteps overlap with each other within errorbars, showing the convergence of PGD attack's success rate and indicating that gradient obfuscation is unlikely the reason for the increased robustness of EBMs.

Figure 4: EP Dynamics

# F EXAMPLES OF IMAGE PERTURBATIONS

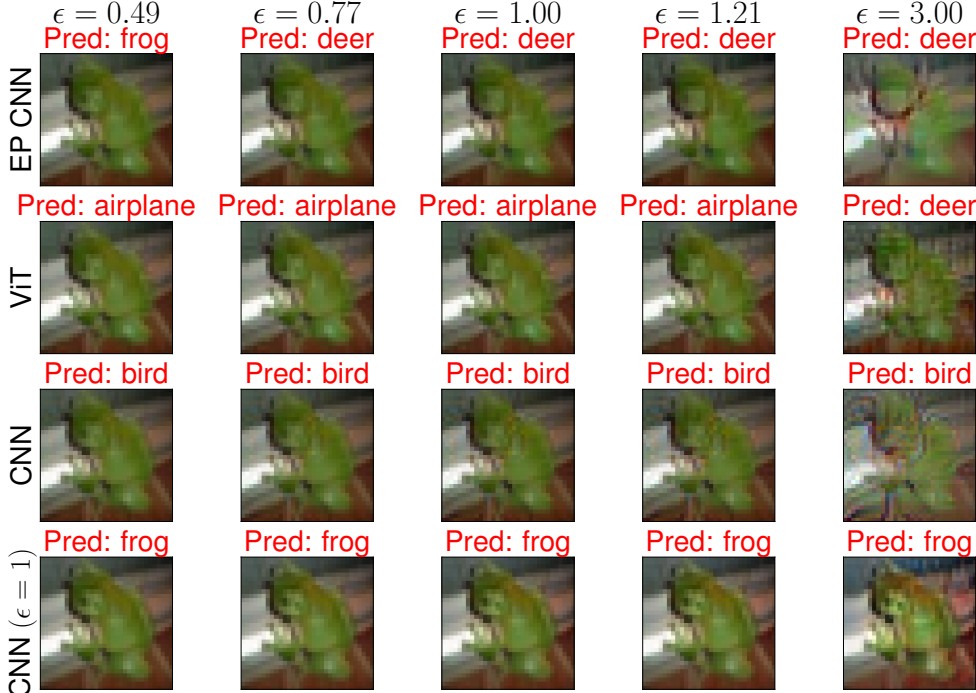

Figure 5: Examples of the $l_2$ perturbations on the attacked images for **BP-CNN**, **AdvCNN** with $l_2$ $\epsilon = 1.00$, **EP-CNN** and **ViT**. Attacks on EBMs/EP are more semantic than those computed on all other models.

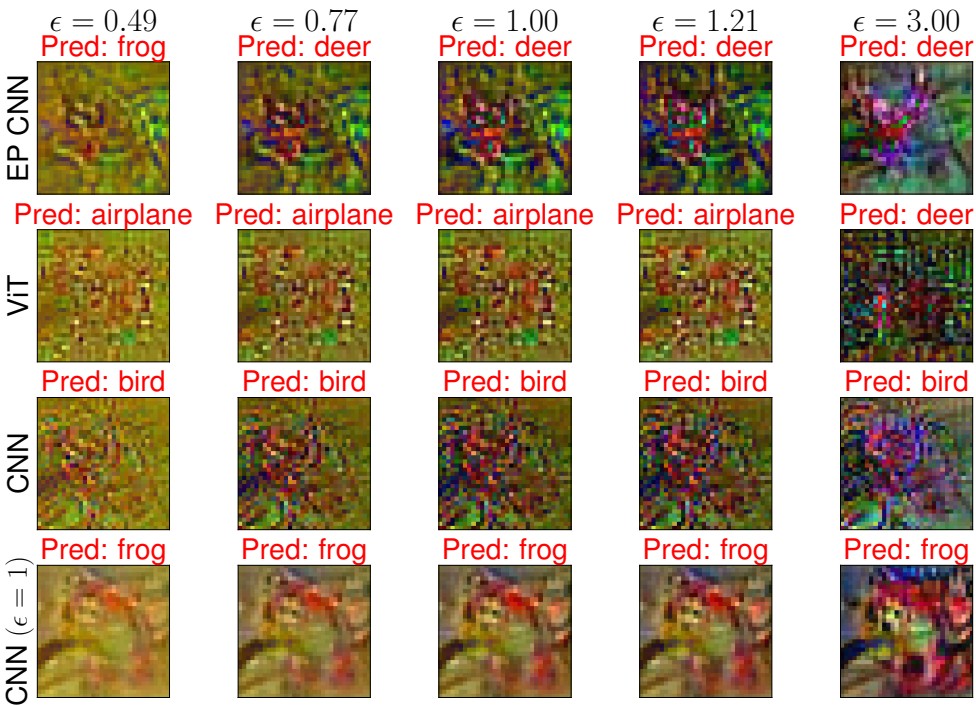

Figure 6: Examples of difference between attacked and clean images for **BP-CNN**, **AdvCNN** trained with $l_2$ $\epsilon = 1.00$, **EP-CNN** and **ViT**. Perturbations on EBMs/EP are more semantic than those computed on all other models.

## G  PGD ATTACK RESULTS

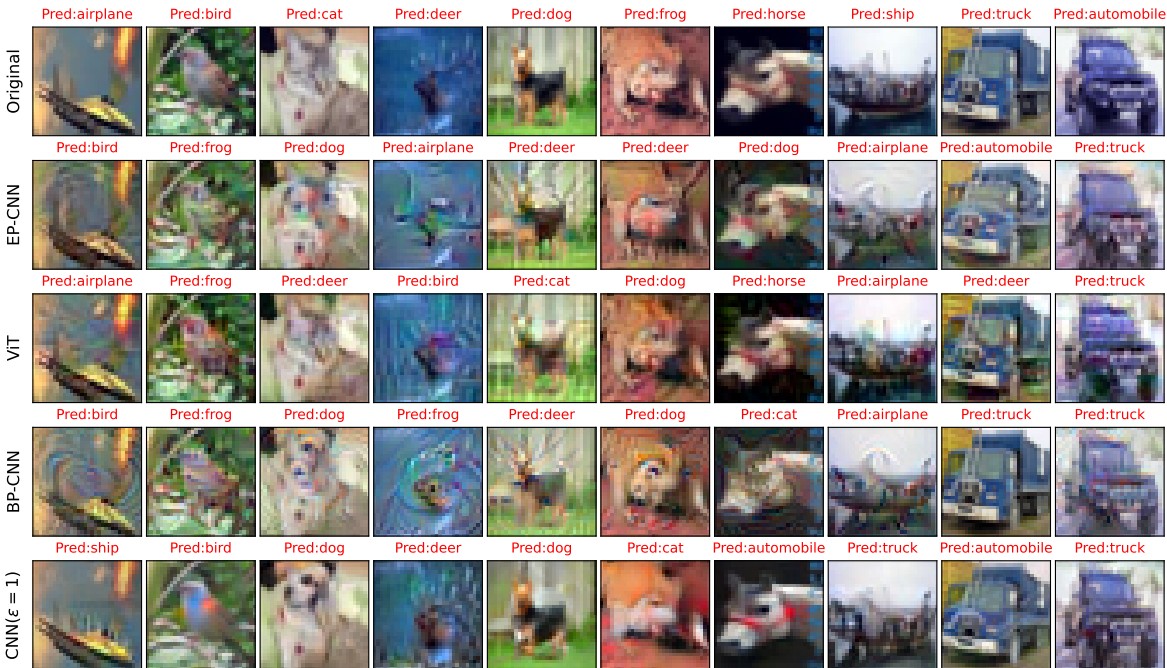

Figure 7: Examples of the $\epsilon = 3.0$ $l_2$ perturbations on images from multiple CIFAR-10 classes for **BP-CNN**, **AdvCNN** trained with $l_2$ $\epsilon = 1.00$, **EP-CNN** and **ViT**.

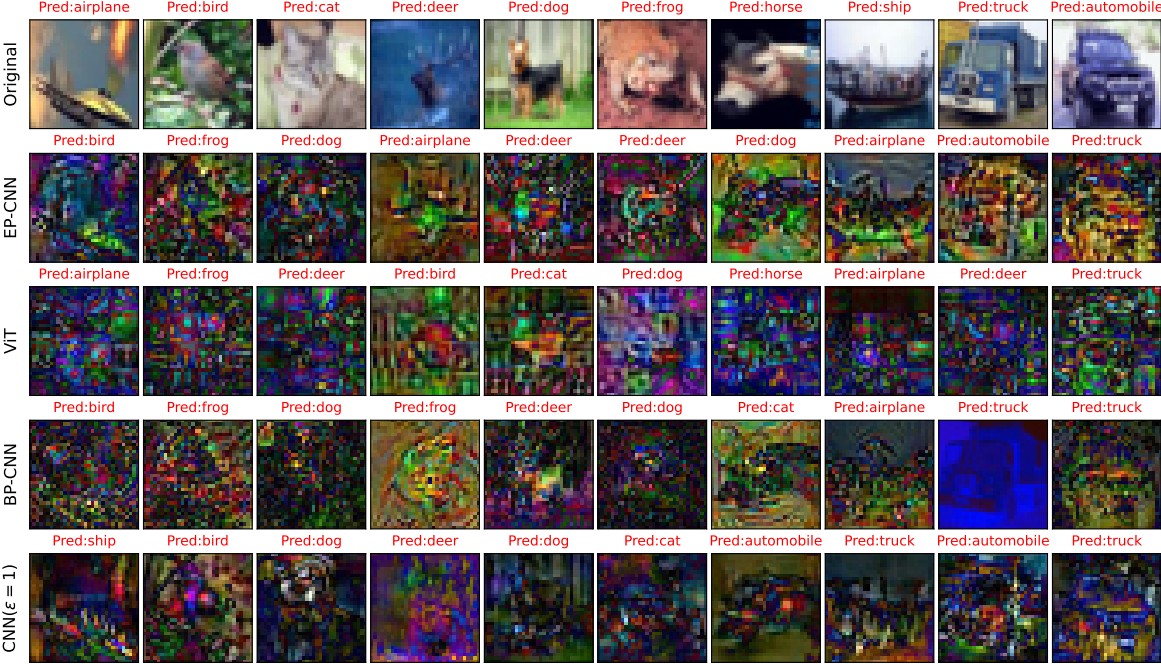

Figure 8: Examples of difference between attacked and clean images belonging to various CIFAR-10 classes for **BP-CNN**, **AdvCNN** trained with $l_2$ $\epsilon = 1.00$, **EP-CNN** and **ViT**.

# H    SQUARE ATTACK RESULTS

# I    C&W ATTACK RESULTS

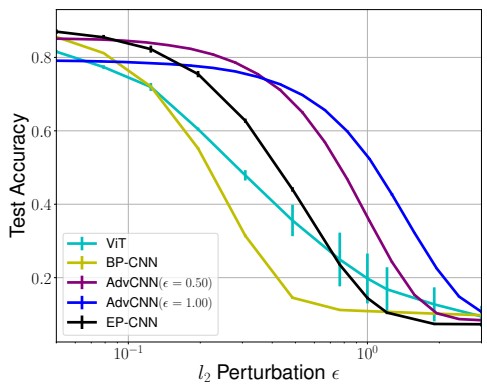 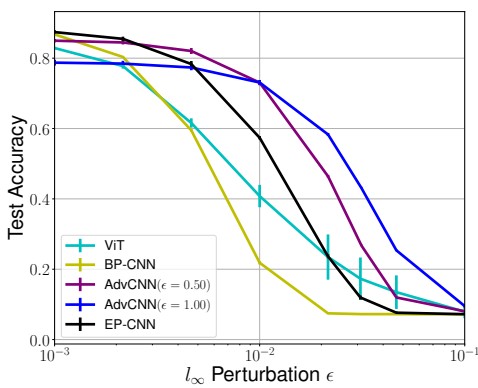

(a) Line graph of accuracy as a function of epsilon in a $l_2$ PGD attack.

(b) Line graph of accuracy as a function of epsilon in a $l_\infty$ PGD attack.

Figure 9: PGD attack results on CIFAR-10 dataset. Lines: **BP-CNN**, **AdvCNN** with $l_2$ $\epsilon = 0.50, 1.00$, **EP-CNN** and **ViT**. Error bars represent the 95% CI over 5 different runs/seeds.

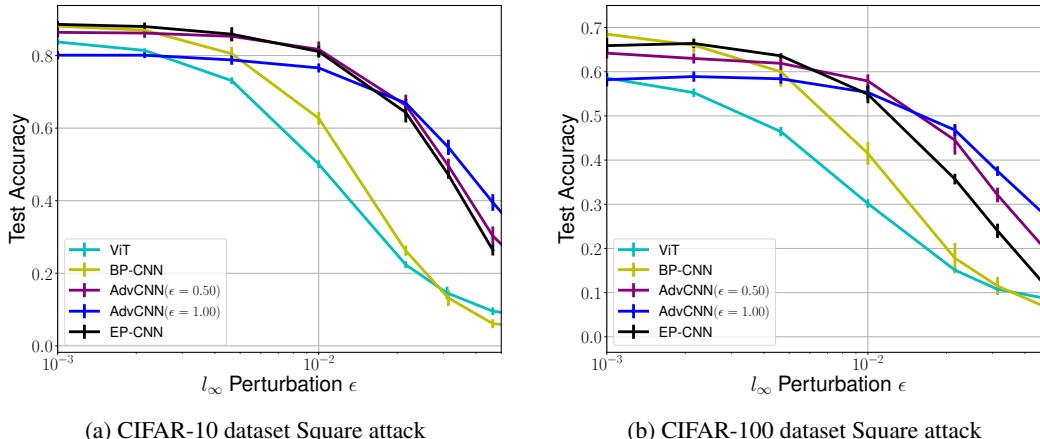

(a) CIFAR-10 dataset Square attack

(b) CIFAR-100 dataset Square attack

Figure 10: Line graph of accuracy as a function of epsilon in a $l_\infty$ Square attack for CIFAR-10 and CIFAR-100 dataset. x-axis: epsilon, y-axis: accuracy. Lines: **BP-CNN**, **AdvCNN** with $l_2$ $\epsilon = 0.50, 1.00$, **EP-CNN** and **ViT**. Error bars represent the 95% CI over 5 different runs/seeds.

## J  CIFAR100 ATTACK RESULTS

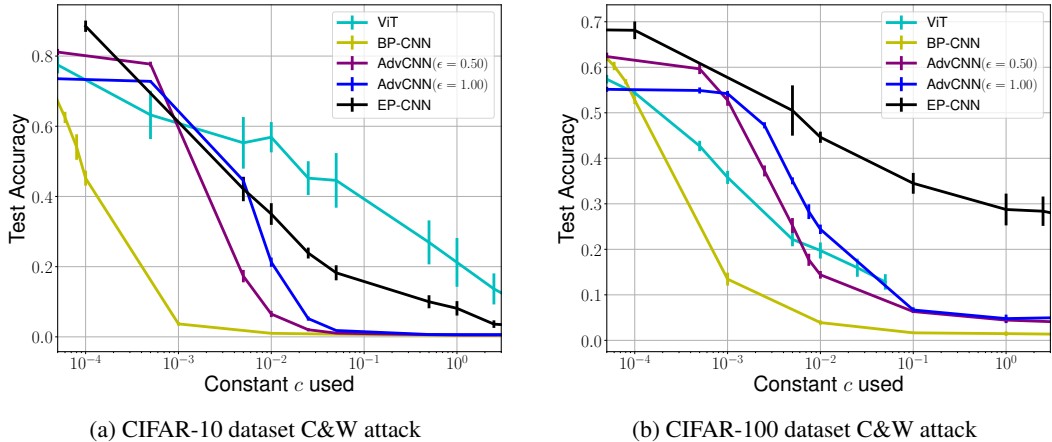

(a) CIFAR-10 dataset C&W attack

(b) CIFAR-100 dataset C&W attack

Figure 11: Line graph of accuracy as a function of perturbation constant c in a $l_2$ C&W attack for CIFAR-10 and CIFAR-100 dataset. x-axis: epsilon, y-axis: accuracy. Lines: **BP-CNN**, **AdvCNN** with $l_2$ $\epsilon = 0.50,\ 1.00$, **EP-CNN** and **ViT**. Error bars represent the 95% CI over 5 different runs/seeds.

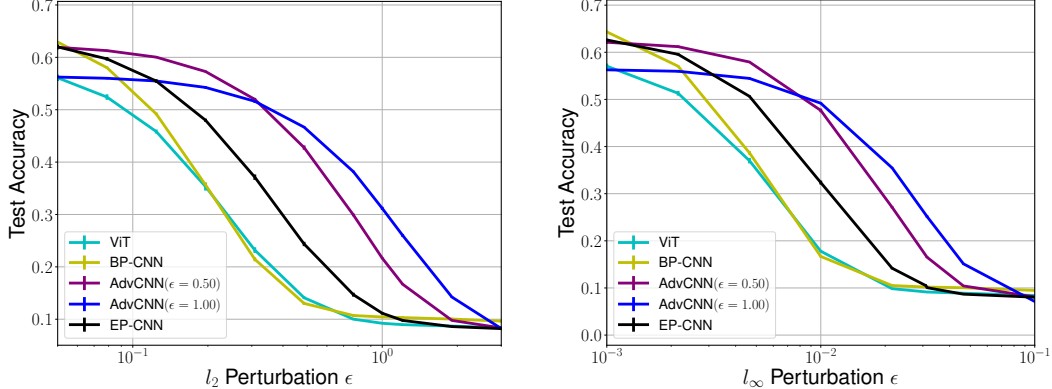

(a) Line graph of accuracy as a function of epsilon in a $l_2$ PGD attack.

(b) Line graph of accuracy as a function of epsilon in a $l_\infty$ PGD attack. x-axis: epsilon, y-axis: accuracy.

Figure 12: CIFAR100 PGD Attacks: Lines: **BP-CNN**, **AdvCNN** with $l_2$ $\epsilon = 0.50,\ 1.00$, **EP-CNN** and **ViT**. Error bars represent the 95% CI over 5 different runs/seeds.

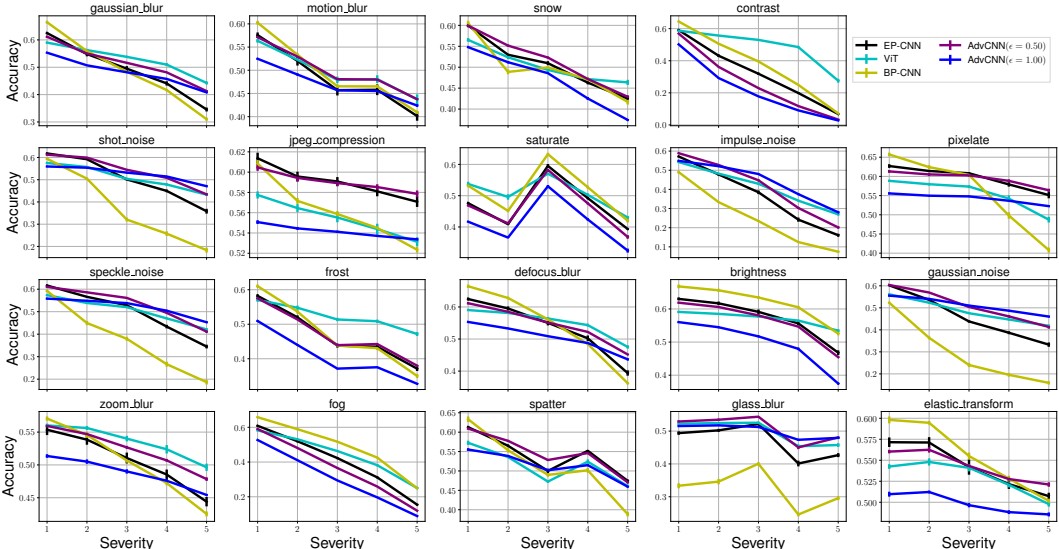

Figure 13: Line graph of accuracy under CIFAR100-C dataset. Lines: **BP-CNN**, adversarially-trained cnn with $l_2$ $\epsilon = 0.50$, $1.00$, **EP-CNN** and **ViT**. x-axis: Corruption Severity (amount of noise added), y-axis: Accuracy. Error bars represent the $95\%$ CI from 5 different runs/seeds.

