# OpenReview forum: "How Robust Are Energy-Based Models Trained With Equilibrium Propagation?"
_ICLR.cc/2024/Conference — Submitted to ICLR 2024_

### Official Review · Reviewer_M88c · 2023-10-29

**Soundness:** 3 good
**Presentation:** 3 good
**Contribution:** 2 fair
**Rating:** 6
**Confidence:** 2

**Summary:**

This paper consider the adversarial robustness of EBM (trained with objective described in (1)), and compares with models trained with classic backpropagation. The results show that EBM are intrinsically more robust, and when we generate adversarial examples that fool the model, the "adversarial examples" also are more meaningful

**Strengths:**

A novel work about evaluating the adversarial robustness of EBM. The work seems pretty thorough,

**Weaknesses:**

First, regarding the adversarial robustness aspect of the paper -- the authors should make clear what the security model is early on. What I mean is that it is not enough to just state the words of "white-box, black-box" and so on, but one needs to have a discussion about what exactly these mean in the EBM case.

For example, it is somewhat clear from the early in the paper that the free phase of the inference may have gradient-masking behavior, but this was never discussed early on (and it is not enough to just say "we did white-box attacks", which in fact is actually already different from previous definitions of "white-box attacks"), and only until quite later in the experiments, we have an actual paragraph of "gradient obfuscation".

In other words, one should lead with a discussion specifically about how we attacked EBM, and be very clear about the inference stage issues. That will make the paper stronger.

Aside from that, while I do like the topic in this paper, the technical content seems to be a somewhat straightforward evaluation of EBM. They are good to know definitely, but the contributions are somewhat still limited.

**Questions:**

No specific question at this point. I have listed my concerns above, to me they are bigger concerns regarding the merits of the paper

---

> ### Author Response · Authors · 2023-11-20
> **Response to Reviewer M88c**
>
> We are grateful to Reviewer M88c for bringing to our notice certain issues. We are also delighted that reviewer M88c finds our paper sound and interesting. Below we include our detailed response.
> # Response to concerns
> 1. The use of the terms ‘white-box’ and ‘black-box’ attacks in the introduction of the paper has been clarified to reflect their significance.
> 2. While we agree that just saying ‘we conducted white-box attacks’ is not enough, we would like to strongly emphasize the fact that we are indeed conducting direct gradient-based attacks, where the adversary has access to the model’s exact gradients across time. This is in contrast to the inexact and indirect gradient-based attacks carried out on Deep Equilibrium Models, which has been discussed in detail in section 2.2. More specifically, we evaluate the gradient of the loss term with respect to the input to compute the perturbation to the images. We thank the reviewer once again for pointing this issue in our writing and we modified section 3.1 and section 3.3 to include details of our attacks, which we hope would improve the readability of the paper.
> 3. There have been several works in the ICLR community that highlight the growing concern of computational complexity of adversarial training, particularly as model size continues to grow. As pointed out in [R1,R2], adversarial training is not just computationally expensive, it often does not generalize to attacks that are different from the ones that the models were trained on. Hence the search for simpler approaches, that are able to generalize their robustness to multiple forms of attacks, is gaining importance. Thus we anticipate that generalized robust models of the future would often be known for their simple implementations, and we hope our work serves as a stepping stone for such models.
>
>
> [R1] Wong, Eric, Leslie Rice, and J. Zico Kolter. "Fast is better than free: Revisiting adversarial training." International Conference on Learning Representations. 2019.
>
> [R2] Shafahi, Ali, et al. "Adversarial training for free!." Advances in Neural Information Processing Systems 32 (2019).

---

### Official Review · Reviewer_GSeC · 2023-10-29

**Soundness:** 3 good
**Presentation:** 2 fair
**Contribution:** 3 good
**Rating:** 6
**Confidence:** 2

**Summary:**

This paper evaluates the robustness of energy-based models (EBMs) via experiments. The authors provide experimental evidence for the EBM robustness against various types of noise. Specifically, EBMs are somewhat robust against both natural noise and adversarial perturbations. The authors first introduce the concept of EBM: some related works and the methodology in particular. Then, the authors present their empirical evaluation of the EBM robustness. Experimental results on the EBM robustness against various natural noise and adversarial examples (both white-box attacks and black-box attacks) are presented.

**Strengths:**

1. As new models are coming out in the machine learning community, it is worth investigating the adversarial robustness of those models.
2. The authors considered many different types of noise: 19 different types of natural noise and two different adversarial attacks (PGD and AutoAttack).
3. The presented result looks to be promising.

**Weaknesses:**

1. The description of how to attack EBM is not detailed enough.
    - If the adversary knows that EBM is used, the adversary will not try using the PGD attack on the traditional loss terms. (If PGD on the traditional loss terms is used for the experiment, it will damage the soundness of this paper.) What is the loss term that is used?
    - Is Temporal Projected Gradient Descent (TPGD) in Appendix 8.2 the attack used for EBM? If so, the authors should present this in the main body of the paper.
2. EBM itself should be described in more detail.
    - How do EBMs make predictions? This detail is needed to understand how to perform gradient-based adversarial attacks on EBM.
    - How costly are EBMs compared to the traditional models? This information is needed to justify the simple model architecture used in experiments.
3. The model architecture (four convolutional layers followed by a fully connected layer) is too simple. Is this because of the expensive computation cost of EBM? How do you know that the findings will generalize to more complex architecture?

**Questions:**

1. How do you measure the severity of each natural noise? The authors should describe how to measure the severity of each type of natural noise.
2. Figure 5 might not be enough to conclude that attacks on EP-CNN are more semantic. (Some people may not agree that the “deer” is induced by the attack.) Probably, it would be better to use a dataset that such semantic changes are more visible. For example, I’m curious about the changes from attacking EP-CNN trained on MNIST or a dataset containing face images.
3. Minor comments
    - To add an appendix, you should use `\appendix` command in LaTeX.
    - Consider making EP-CNN results more visible in the plot. For example, make other lines thinner in the plot, so that you can emphasize EP-CNN results. Also, it is better to render EP-CNN results after rendering other lines so that the other lines are now drawn over the EP-CNN results.

---

> ### Author Response · Authors · 2023-11-20
> **Response to Reviewer GSeC**
>
> We thank Reviewer GSeC for their detailed feedback and their critical assessment of the paper. Here we address reviewer GSeC’s concerns in detail.
>
> # Response to concerns
>
> 1. We agree with the reviewer that it is unclear what loss function is used to train and attack the EP model, and that this information is important for accurately assessing the robustness of EP to adversarial attacks. Cross-entropy loss is used to train and attack all models tested here, including EP. As a result, we believe the experiments are sound because we are training and attacking EP the same way as all other models, which is the standard way to train and attack deep neural network classifiers in general. We will add these details in the main text to ensure that this is clear.
> 2. We admit that details on carrying out white-box attacks are not straightforward and are grateful to reviewer GSeC for pointing this out. As a result, we are moving the section on temporal projected gradient descent from the appendix to the methods section 3.2.
> 3. Describing EBM in detail
>     - We are including clarifying sections in the methods section which explains how predictions are made by models trained with Equilibrium Propagation.
>     - The training cost of such models are discussed in section 4.2. As of the time of writing this paper, EP has only been able to scale to ImageNet32 [R1] and takes exponentially more time and computational resources to attack, when implemented on standard GPU based hardware. Our implementations offer a speedup of 30% as discussed in section 4.2, which was one of the major reasons we were able to carry out gradient-based attacks in a reasonable amount of time.
> 4. The implemented model architecture is kept simple because it is expensive to train on standard hardware. As the world starts moving towards non-standard spiking hardware, such models will become the more efficient alternative. We are confident that these would scale to more complex architectures / problems as previous work by Laborieux et al. [R1] have shown that even such small models perform well across various complex tasks (CIFAR 10, CIFAR100, and ImageNet-32).
>
> # Answers to Questions by Reviewer GSeC
> 1. Severity of natural noise is described in the paper that introduced the CIFAR10-C dataset [R2], which is a standard benchmark used to evaluate robustness to natural noise.
> 2. We admit to the fact that attacked images in Figure 5 may not always appear as “deer” to our readers. To this end, we are including a collage of images from multiple classes bearing perturbations in the appendix, to convince our readers that perturbations in EP are not limited to the background.
> 3. Minor comments
>     - We apologize for the use of sections instead of using /appendix in the paper. We thank reviewer GSeC for pointing out this error. We have fixed this issue in our paper.
>     - We are grateful to reviewer GSeC for pointing out the fact that the plots of EP should be rendered above the lines corresponding to robust accuracies of other models.
>
>
> [R1] Laborieux, Axel, and Friedemann Zenke. "Holomorphic equilibrium propagation computes exact gradients through finite size oscillations." Advances in Neural Information Processing Systems 35 (2022): 12950-12963.
>
> [R2] Hendrycks, Dan, and Thomas Dietterich. "Benchmarking Neural Network Robustness to Common Corruptions and Perturbations." International Conference on Learning Representations. 2018.

---

### Official Review · Reviewer_8aDx · 2023-10-29

**Soundness:** 3 good
**Presentation:** 2 fair
**Contribution:** 2 fair
**Rating:** 6
**Confidence:** 4

**Summary:**

This paper empirically studies the energy based models especially the Equilibrium Propagation CNN (EP-CNN) in terms of adversarial robustness and natural noise on CIFAR. The results show that the EP-CNN is robust compared to ViT and adversarially trained CNN.

**Strengths:**

It’s an interesting direction to look into the robustness for EP models and the paper shows some interesting preliminary results on CIFAR.

The paper is educative and interesting to read through, could possibly lead to many followups.

**Weaknesses:**

The paper claims that EP model is significantly more robust. However, the improvements are kind of marginal according to the line graphs in fig 2-4. Please quantify the improvements with specific numbers if the authors want to claim the significance.

The novelty is limited. This paper is mostly an empirical investigation on existing model architectures though I appreciate the new angle and experimental efforts.

The paper did not clearly state the novelty in Section3 Methods -  Are there any novelty proposed in this section or mainly repeat the approaches in existing works? If so it’s better to change the section title and make this clear.

It would be more convincing if the authors can theoretically prove or provide more solid insights why EP models are more robust.

EP models are harder to train and converge as discussed in Limitation, limiting the approach to scale up. It would be helpful to explore the benefits of EP models on other tasks besides image classification where they can show advantages over ViT and regular CNNs.

[post rebuttal]
Thank the authors for the response and additional results. Some of my concerns are addressed, while my last two concerns still remain. I am happy to raise my score to 6.

**Questions:**

See weakness above.

---

> ### Author Response · Authors · 2023-11-20
> **Response to Reviewer 8aDx**
>
> We thank Reviewer 8aDx for their detailed feedback and their critical assessment of the paper. Here we address reviewer 8aDx’s concerns in detail.
> # Response to concerns
> 1. We added a table summarizing the results of all adversarial attacks against all models tested. Results for EP are presented separately, at the end of the table, for each of the attacks. The best performing robust accuracies for each epsilon has been highlighted and comparison with EP shows that EP performs as good as the adversarially trained models for moderate attacks.
>
> 2. We would like to address the concerns about novelty in terms of our approach.Energy-based models are a class of AI models that learn through local learning rules and the limited previous work that exists, has only focused on different training techniques and architectures but has not addressed the robustness of these models. We were the first to show the inherent robustness of models trained with Equilibrium Propagation, through an ensemble of attacks where the gradient-based attacks were carried out exactly. As a result, we believe the novelty and contribution of the results is more than marginal.
>
> 3. We have added detailed procedure on attacking the EBM model and renamed section 3.1 to clearly reflect our contributions
>
> 4. We add an additional section into the appendix of the paper where we provide an intuitive discussion of where the robustness comes from (energy landscape of the EBM). We leave thorough theoretical analysis for future work.
>
> 5. We do not anticipate our results would change very much under different data modalities. However, this is outside of the scope of this work, which was to test the robustness of existing EP models (which have been applied only to image classification in previous works). We do agree that this is a good suggestion for future work, though.
>
>
> We thank you again for your valuable feedback.  Please let us know if the above explanations do not address your concerns completely. We are happy to answer any further questions.

---

### Official Review · Reviewer_QZD5 · 2023-10-30

**Soundness:** 2 fair
**Presentation:** 2 fair
**Contribution:** 3 good
**Rating:** 6
**Confidence:** 3

**Summary:**

This paper analyzes the robustness of the Energy-Based Models (EBMs) trained with Equilibrium Propagation (PB) under natural corruptions and adversarial attacks. The experiments mainly focus on the PGD attacks, taking CIFAR-10 and CIFAR-100 as the datasets. There are three baseline models from different categories, namely, BP-CNN, AdvCNN, and ViT. Besides, at the end of the article, this paper discusses its limitations and future directions.

**Strengths:**

1. This paper is the pioneer in investigating the robustness of EBMs trained with EP under natural corruption and adversarial attacks, which may spark new insights and follow-up works.
2. This paper honestly discusses its limitations at present and provides several future directions.

**Weaknesses:**

1. Though the paper mentions that the high time cost limits EP models to relatively shallow architectures, I think the experiments that mainly focus on PGD are not enough. There are other attack methods, e.g., FGSM [1], C&W [2], AA [3], etc. The conclusion that EBMs trained with EP have better robustness will be more convincing if more attack methods are evaluated, even though the models have relatively simple architecture and the datasets are relatively small.
2. The superiority of the EP models in robustness is mainly shown by experimental results, which lack theoretical analysis and support. If rigorous mathematical analysis is too hard, then at least an intuitive analysis based on formulas that show why EP models can function well should be provided.

---
[1] Goodfellow, I. J.; Shlens, J.; and Szegedy, C. 2014. Explaining and harnessing adversarial examples. arXiv preprint arXiv:1412.6572.

[2] Athalye, A.; Carlini, N.; and Wagner, D. 2018. Obfuscated gradients give a false sense of security: Circumventing defenses to adversarial examples. In International conference on machine learning, 274–283. PMLR.

[3] Croce, F.; and Hein, M. 2020. Reliable evaluation of adversarial robustness with an ensemble of diverse parameter-free attacks. In International conference on machine learning, 2206–2216. PMLR.

**Questions:**

Including the concerns and questions mentioned in Weaknesses, I still have the following questions:
1. For the second term on the right side of Equation 1, I think it is possible that there are pooling layers following the FC layers. In other words, the second term can still contain $\mathcal{P}(\cdot)$?
2. According to my understanding, in Section 3.1, the training goal is to update the convolution weights $w_{n}$? However, I am unclear how the weights $w_n$ are updated based on Equations 4-6 related to the states $s^{n-1}$ and $s^{n+1}$. Can there be more explanation?
3. As mentioned in the limitation, ``EP is relatively sensitive to the hyperparameters used to train the model as well as the seed used to initialize the weights``. Are there some heuristic rules to set the hyperparameters and the weights' initialization?
4. Some editorial issues (just listed a few):
- Better not to use contractions such as ``didn't``, ``don't``, ``weren't``.
- It is a bit strange that there are the right brackets after BP-CNN and AdvCNN, respectively, but not the right bracket after ViT in Section 4.1.1.
- A missing period ``.`` after the second paragraph in Section 4.1.1.
- Inconsistent representations when referring to figures. For example, ``Fig. 2 and Fig. 10`` and ``Figure 1`` on page 8.
- On page 6, ``Figure 3,4`` should be ``Figures 3,4``.
- ...

---

> ### Author Response · Authors · 2023-11-20
> **Response to Reviewer QZD5**
>
> We thank Reviewer QZD5 for their insightful feedback and their efforts in pointing out the grammatical issues. We are encouraged that Reviewer QZD5 finds our approach pioneering and capable of leading to multiple follow-up works, this is indeed the kind of reception we hoped for. Below we address Reviewer QZD5's concerns in detail and provide additional experimental results.
>  # Responses to concerns
> 1. We have indeed conducted the AA attack and presented our results in Figure 4 in the original submission. We also conducted C&W attacks across all models and have added the results in the appendix. However, we would like to discuss a few details on FGSM and C&W.
>    - FGSM is a 1-step version of PGD and thus superseded by both PGD and AA, which we use in other experiments and have presented in the results section of the original submission. Many other previous works have also illustrated that PGD attacks provide a much better test of a model’s robustness relative to FGSM. For all of these reasons, we believe it is unnecessary to include FGSM attacks in our experiments.
>     - We agree with the reviewer that the inclusion of C&W attacks would provide a more thorough evaluation of EP’s robustness, and we thank the reviewer for this suggestion. We have included results from a C&W attack in the modified submission. Under these C&W attacks, EP is one of the most robust models, even more than the adversarially trained models, for the CIFAR-100 dataset.
> 2. We appreciate your comment about our work lacking theoretical analysis. We added a section in the appendix that provides intuitive analysis and hypothesis of why EBM models are more resistant to adversarial attacks as opposed to traditional training methods and we leave a thorough theoretical approach for future work.
> # Answers to questions by Reviewer QZD5
> 1. There are no pooling layers following the fully connected layers, so the second term on the right side does not contain any pooling terms.
> 2. Weight update rules are provided in the supplementary section of the original paper[R1].
> 3. For a hyperparameter search, one has to check the state of the system after activations have been applied and make sure the value of the states post-activation are not zeroed.
> 4. We are very grateful to you for pointing out the editorial issues and we have made sure that the paper is grammatically consistent and easier to read.
>
>
> Thank you again for your time and effort in reviewing our paper! We believe your suggestions have strengthened our paper greatly. Please let us know if the above explanations do not address your concerns completely. We are happy to answer any further questions.
>
>
> [R1] Laborieux, Axel, et al. "Scaling equilibrium propagation to deep convnets by drastically reducing its gradient estimator bias." Frontiers in neuroscience 15 (2021): 633674.

---

### Author Response · Authors · 2023-11-20
**We are grateful for the reviews!**

We thank all the reviewers for their valuable feedback and their constructive comments. We are thrilled that the reviewers find our approach pioneering (**QZD5, 8aDx, M88c**), our experiments thorough (**GSeC, M88c**) and our paper well-written and honest in its claims (**QZD5, 8aDx**).

We have addressed individual questions of reviewers in separate responses. In the revised version, we incorporated all reviewers' suggestions by adding more theoretical insights, clarifying the experimental details, more experimental results in the appendix, as well as more implementation details. Here we briefly outline the updates to the revised submission for the reference of reviewers.
**Paper Updates**:

**[Section 3.1 Existing Equilibrium Propagation Framework and Temporal Projected Gradient Descent]** We clarified the language on introducing the equilibrium propagation model (**8aDx**) and moved the details of the attack to the main body of the text (**GSeC, M88c**)

**[Updated Figures]** We updated figures such that the robust accuracy curves for EP-CNNs are rendered after curves of other models (**GSeC**)

**[Appendix More Attacks]** We included results of C&W attacks across all models and for both datasets presented in this paper. (**QZD5**)

**[Appendix Table of robust accuracy across all models]** We included a table containing robust accuracies of all models for selected epsilons, highlighting the attacks where EP-CNN outperforms other models (**8aDx**)

[**Editorial Issues**] We fixed grammatical errors and made the paper more consistent which improves the readability (**QZD5**)

---

### Meta-Review · Area_Chair_4JRJ · 2023-12-05

**Metareview:**

This paper investigates the robustness of energy-based models (EBMs) to both natural corruptions and adversarial attacks. The authors demonstrate that EBMs exhibit greater robustness than transformers and comparable robustness to adversarially-trained deep neural networks (DNNs) across white-box, black-box, and natural perturbations. This is achieved without compromising clean accuracy and without the need for adversarial training or additional techniques. The paper is well-organized and interesting. However, a weakness of the paper is the absence of theoretical analysis and the limitation of testing the model on more complex tasks, resulting in an insufficient justification for the proposed method. The rebuttal partially addresses the concerns raised by reviewers, leading to an overall borderline rating. During the discussion, no reviewer strongly supports the paper. The Area Chair acknowledges the paper's promise but suggests that the experiments need improvement to better justify and analyze the proposed method. Considering these factors, the Area Chair recommends rejecting the paper at the current stage. The AC encourages the authors to enhance their paper by incorporating the valuable suggestions provided by the reviewers and to resubmit the revised paper to the next venue.

**Justification For Why Not Higher Score:**

This paper lacks strong empirical results or theoretical analysis to justify the proposed method. The rebuttal did not adequately address the concerns raided by the reviewers. The AC recommends rejecting the paper.

**Justification For Why Not Lower Score:**

NA

---

### Decision · Program_Chairs · 2024-01-16

Reject